# Genetic Diversity and Population Structure of a Rhodes Grass (*Chloris gayana*) Collection

**DOI:** 10.3390/genes12081233

**Published:** 2021-08-10

**Authors:** Alemayehu Teressa Negawo, Meki S. Muktar, Yilikal Assefa, Jean Hanson, Alieu M. Sartie, Ermias Habte, Chris S. Jones

**Affiliations:** 1Feed and Forage Development Program, International Livestock Research Institute, Addis Ababa P.O. Box 5689, Ethiopia; A.Teressa@cgiar.org (A.T.N.); M.Shehabu@cgiar.org (M.S.M.); Y.Assefa@cgiar.org (Y.A.); jeanhanson2010@gmail.com (J.H.); sartiealieu@hotmail.com (A.M.S.); e.habte@cgiar.org (E.H.); 2The Pacific Community (SPC), Private Mail Bag, Suva, Fiji; 3Feed and Forage Development Program, International Livestock Research Institute, Nairobi 00100, Kenya

**Keywords:** DArTSeq markers, genetic diversity, Rhodes grass (*Chloris gayana*), subset

## Abstract

Rhodes grass (*Chloris gayana* Kunth) is one of the most important forage grasses used throughout the tropical and subtropical regions of the world. Enhancing the conservation and use of genetic resources requires the development of knowledge and understanding about the existing global diversity of the species. In this study, 104 Rhodes grass accessions, held in trust in the ILRI forage genebank, were characterized using DArTSeq markers to evaluate the genetic diversity and population structure, and to develop representative subsets, of the collection. The genotyping produced 193,988 SNP and 142,522 SilicoDArT markers with an average polymorphic information content of 0.18 and 0.26, respectively. Hierarchical clustering using selected informative markers showed the presence of two and three main clusters using SNP and SilicoDArT markers, respectively, with a cophenetic correction coefficient of 82%. Bayesian population structure analysis also showed the presence of two main subpopulations using both marker types indicating the existence of significant genetic variation in the collection. A representative subset, containing 21 accessions from diverse origins, was developed using the SNP markers. In general, the results revealed substantial genetic diversity in the Rhodes grass collection, and the generated molecular information, together with the developed subset, should help enhance the management, use and improvement of Rhodes grass germplasm in the future.

## 1. Introduction

Rhodes grass (*Chloris gayana* Kunth) is an important tropical C4 grass widely used throughout the tropical and subtropical regions of the world [1,2,3]. It is either an annual or perennial, high yielding and good quality forage grass that is also used as a cover crop to improve soil fertility and reduce soil nematodes [1,2].

Rhodes grass is a primarily cross-pollinated diploid or tetraploid (with a basic chromosome number, x = 10) highly polymorphic forage grass species [2]. It has a deep root system and can withstand extended periods of drought [1], and grows in a wide range of ecologies and soil types [1,2] with no known economically important biotic stressor [2]. It is reported to be a salt-excluding halophyte that secretes excess salts transported into the leaves [4] and this characteristic makes this grass species one of the most important candidate forages for economic utilization in saline environments such as in saline affected irrigation farming [4,5,6]. Several cultivars with improved performance, particularly in drought and low temperature prone areas, have been developed and commercialized [2]. A few cultivars have also been reported to be frost tolerant [1,2]. Diploid cultivars have been reported to be more resistant/tolerant to drought, salt, low temperatures, pests and diseases than the higher producing tetraploid cultivars [1,7,8]. For example, tetraploids are more susceptible to root-knot nematode than diploids [7]. In general, Rhodes grass has some good agronomic and morphological characteristics that make it a resilient forage crop to consider under the current dynamics of climate change to increase the availability of feed resources for sustainable livestock production.

Conventional molecular marker technologies, such as inter simple sequence repeats (ISSR), amplified fragment length polymorphisms (AFLP), sequence-related amplified polymorphisms (SRAP) and restriction fragment length polymorphisms (RFLP), have been used to study a few known Rhodes grass cultivars [8,9,10,11,12,13], although the results of these previous studies were limited in their abiltiy to provide insight into the genetic diversity of the species, both by the number of genotypes included and the low density markers used. Despite its potential economic importance in drought and saline affected environments and the fact that it is a genetically polymorphic species, genomic studies have been limited, making Rhodes grass one of the orphan crops in the application of molecular genetics. This has constrained the opportunity for further use of germplasm and enhanced genetic gains through molecular breeding and related technologies. The International Livestock Research Institute (ILRI) maintains 106 Rhodes grass accessions in its genebank in Addis Ababa, Ethiopia [14]. Besides ILRI, there are other significant collections; for example, the Genetic Resources Research Institute (GeRRI) (Kenya), United States Department of Agriculture (USDA), International Center for Biosaline Agriculture (ICBA) (United Arab Emirates, UAE),and Australian Pastures Genebank (APG, Australia) hold 1196, 132, 116 and 103 accessions, respectively [14]. The Rhodes grass accessions in the ILRI collection were collected from different parts of Africa and from India and contain some improved cultivars. With the exception of some basic characterization data collected during routine field conservation operations, the collection has not been well studied in the past, and thus little is known about the genetic diversity, population structure or genomic information available in the collection. The only reported study was performed by Ponsens and colleagues, who indicated the presence of genetic variation in the collection based on agro-morphological traits [15]. Here, we report on the development of genomic information, which reveals the genetic diversity and population structure of the collection, and identify a subset using the genotyping-by-sequencing (GBS) approach of the DArTSeq platform. 

## 2. Materials and Methods

### 2.1. Materials

A total of 104 Rhodes grass accessions, maintained in the ILRI forage genebank, were used in this study. Leaf samples could not be obtained for two accessions, as they were not in the field at the time of this study. These accessions were collected from 14 African countries and one from India (Figure 1, Appendix A). The origin of six accessions was unknown. 

### 2.2. DNA Extraction and Genotyping

Leaf samples were collected from plants growing in the ILRI Zwai field Genebank site, Oromia, Ethiopia. DNA was extracted from freeze dried leaf samples using a DNeasy Plant Mini kit (Cat No./ID:69106) according to the manufacturer’s instruction. DNA quality and quantity were assessed using a DeNovix DS-11 spectrophotometer. DNA samples were diluted to a concentration of 50–100 ng/µL, and 25 µL of the diluted DNA samples was aliquoted into 96-well fully skirted plates, packed, and shipped for genotyping.

Genotyping was performed using the DArTSeq platform at Diversity Arrays, Canberra, Australia. The DArTSeq (Single nucleotide polymorphism (SNP) and SilicoDArT) marker data were generated according to the DArTSeq protocol as described elsewhere [16]. The sequence fragments of the generated markers were aligned onto the reference genomes of closely related species. The reference genomes were selected from public genome databases based on their taxonomic relationship with Rhodes grass. 

### 2.3. Data Analysis 

The genotyping data were analyzed using different R software (https://www.r-project.org/, accessed on 16 January 2019) packages. Data missing percentage, allele frequency, polymorphic information content (PIC) and genetic distance were calculated using the R base function. The fragment length of the markers was summarized using the stringr R package [17]. The PIC value was calculated using the formula PIC = 1 − ∑p_i_^2^ where P_i_ is the frequency of alleles at each locus [18]. The markers were filtered first for missing percentage (≤20%) and then for the level of informativeness (PIC ≥ 0.2) using a locally written script in the R statistical software. To study the genetic relatedness among the accessions, Euclidean genetic distance and hierarchical cluster (hclust object) were calculated using the dist () and hclust () functions, respectively, in the R software. The hclust objects were converted into dendrogram objects using the R package dendextend [19] and the phylogenetic tree was visualized using the plot () function of the R software. The dendrograms from the SNP and SilicoDArT markers were visualized side by side using the tanglegram function while the cophenetic correlation coefficient between the dendrograms was calculated using the cor-cophenetic function of dendextend [19]. 

The optimal number of clusters was determined using the find.clusters function of the R package adegenet [20]. The discriminant analysis of principal components (DAPC) function of adegenet [20] was used to select the top 1000 markers contributing to the population diversity and to infer cluster membership probability and assign individual accessions into the different clusters. The cluster groups were visualized using the fviz_cluster function of the R package factoextra [21]. Analysis of molecular variance (AMOVA) was conducted using GenAlEx software [22] with 9999 permutations to partition the genetic variations into between and within clusters. 

Population structure was analyzed using the admixture model in STRUCTURE software [23,24], and the probability of 2 to 20 subpopulations (K) was estimated using the admixture model, 100,000 Markov Chain Monte Carlo (MCMC) repetitions and a 100,000 burn-in period. The result of the run was uploaded online to the software “STRUCTURE HARVESTER” [25], and the optimal number of subpopulations was determined using the Evanno delta K method [26]. AMOVA was used to partition the genetic variation into among and within subpopulation variations using the GenAlEx software with 9999 permutations [22].

Finally, the 1000 selected SNP markers were used to develop a subset using the R package Core Hunter v.3.1 [27]. The subset contained 20% of the collection representing the maximum diversity of the collection. AMOVA was used to assess the representativeness of the developed subset using GenAlEx 6.5 software [22].

## 3. Results

### 3.1. Informativeness and Diversity of the Markers

Genotyping data of 193,988 SNP and 142,522 SilicoDArT markers was generated for 104 Rhodes grass accessions. After filtering the data with missing percentage (≤20%) and PIC (≥0.2), 30,279 and 59,164 SNP and SilicoDArT markers, respectively, were retained for preliminary analysis to select the top 1000 markers contributing to genetic differentiation among the accessions. The PIC values ranged from 0.01 to 0.50 and 0.02 to 0.50 with an average of 0.18 and 0.26 for SNP and SilicoDArT markers, respectively (Figure 2). A large number of the markers had a low PIC (<0.2), which indicates the high frequency of occurrence of one of the alleles. 

The markers’ sequence length ranged from 20–69 base pairs (bp) with an average of 65 bp and 63 bp for SNP and SilicoDArT markers, respectively. Over 80 and 78% of the SNP and SilicoDArT markers, respectively, had a fragment length of 69 bp. Among the SNP variations, transitions (57.3%) were more frequent than transversions (42.7%) (Figure 3). The proportion of polymorphisms due to G/A (15.74%) and C/T (15.44%) transitions and T/C (13.03%) and A/G (13.06%) transitions were similar. The proportion of polymorphisms due to different transversions ranged from 4.81% to 5.95%. 

### 3.2. Genome Mapping of the DArTSeq Markers onto Reference Genomes and Selection of Genome-Wide Representative Markers 

Rhodes grass belongs to the subfamily *Chloridoideae* of the *Poaceae* grass family. In the *Chloridiodeae* subfamily, the reference genomes of four grass species, namely tef (*Eragrostis tef*) (GenBank: GCA_000970635.1) [28], Manila grass (*Zoysia matrella*) (GenBank: GCA_001602295.1) [29], and finger millet (*Eleusine coracana*) (GenBank: GCA_002180455.1) [30] are available. All of these reference genomes are yet to be assembled to the chromosome level. Based on the taxonomic classification, the reference genomes of tef, finger millet and Manila grass were selected as good candidates to map the genome-wide distribution of the generated DArTSeq markers. In addition, the reference genome of foxtail millet (*Setaria italica*) (GenBank: GCF_000263155.2) [31,32], a model species for grass genomic studies, was used to map the genome distribution of the markers. Table 1 shows the number of markers mapped onto the selected reference genomes. Relatively more markers were mapped onto the finger millet genome followed by the tef genome. The least number of markers mapped onto the foxtail millet genome. 

### 3.3. Genetic Diversity and Relationship of the Rhodes Grass Collection

A preliminary analysis was conducted using markers containing ≤20% missing data and PIC values of ≥0.2 using the DAPC function of the R package adegenet. Based on the loading score, the top 1000 markers contributing to the diversity were selected for further in-depth analyses. Accordingly, the selected markers had an average PIC value of 0.37 and 0.36 for the SNP and SilicoDArT markers, respectively.

Using the genetic dissimilarity matrix, the genetic relationship between accessions was analyzed based on hierarchical clustering. Figure 4 (and Appendix A) shows the hierarchical clustering of the collection. Using the SNP markers, the collection was grouped into two main clusters while the collection was grouped into three clusters using SilicoDArT markers. A strong cophenetic correlation of 82% was observed between hierarchical clustering generated using the SNP markers and the SilicoDArT markers. The main clusters were further subdivided into sub-clusters for both marker types. As revealed by Mantel correlation analysis, no clear correlation was observed between the hierarchical clustering and the geographic origin of the accessions (*r*= 0.0002, *p*-value = 0.4845 for silicoDArT and *r* = −0.0521, *p*-value = 0.8504 for SNP markers).

### 3.4. Cluster Analysis of the Collection

For population cluster analysis, the optimum number of clusters and cluster membership of each accession were estimated using the ‘find.clusters’ and ‘DAPC’ functions of adegenet [20]. The fviz_cluster function of the R package factoextra [21] was used to visualize the cluster plot. The suggested optimal number of clusters were two and three for SNP and SilicoDArT markers, respectively. Figure 5 shows the cluster plots based on the estimated number of clusters. For the SNP markers, the first and second dimensions accounted for 25.9% of the total genetic variation and grouped the accessions into two main groups (Figure 5a). The first cluster contained 77 accessions while the second cluster contained 27 accessions. In the case of the SilicoDArT markers, the first two dimensions accounted for 20.5% of the total genetic variation and grouped the accessions into three clusters, each cluster containing 10, 23 and 71 accessions, respectively (Figure 5b). In line with the cluster analysis, the accessions were assigned to the different clusters with clear membership probability (Figure 6). Analysis of molecular variance (AMOVA) showed that the ‘among clusters’ variation accounted for 24% and 19% of the total genetic diversity and significance while the ‘within cluster’ variation accounted for 76% and 81% of the total genetic diversity for SNP and SilicoDArT markers, respectively (Table 2). 

### 3.5. Population Structure Analysis of the Germplasm Collection 

The population structure analysis based on Bayesian inference was undertaken using the software STRUCTURE, and the delta K result suggested the presence of two main subpopulations using both marker types (Figure 7). Based on the SNP markers, the first subpopulation contained 30 accessions while the second subpopulation contained 74 accessions originating from different countries. There was a second peak at K = 6, suggesting a further subgrouping of the subpopulations. Similarly, using the SilicoDArT markers, the first subpopulation contained 28 accessions while the second subpopulation contained 76 accessions. Table 3 shows the AMOVA result partitioning the total variation into among and within subpopulation variations. The among subpopulation variation contributed 30% and 19% (for SNP and SilicoDArT markers, respectively) of the total genetic variation while the within subpopulation variation contributed 70% and 81% (for SNP and SilicoDArT markers, respectively) of the total genetic variation in the population structure.

### 3.6. Subset Development 

A subset containing representative diversity of the collection was developed using SNP markers (Table 4). The subset contained 21 accessions, which originated from different countries such as Congo, Ethiopia, India, Kenya, Malawi, Tanzania, South Africa, Zimbabwe and one accession of unknown origin. The AMOVA result (Table 5) showed that there was no significant difference between the developed subset and the rest of the population in terms of genetic diversity. The within groups diversity contributed almost 100% the genetic variation. 

## 4. Discussion

### 4.1. Genotyping and Marker Diversity 

Here, we report on the development of high throughput DArTSeq markers for a Rhodes grass collection. Large numbers of SNP and SilicoDArT markers were generated and used to assess the genetic diversity and population structure in the collection. The generated SNP and SilicoDArT markers had an average PIC value of 0.18 and 0.26, respectively. To our knowledge, this is the first report of its kind in Rhodes grass, both in terms of the number of markers as well as the number of accessions studied. 

The SNP data were used to assess the different types of variation, i.e., transitions and transversions. The results revealed that the proportion of SNP variation due to transitions (57.3%) was higher than for the transversions (42.7%). Similar results have been reported in other plant species including Arabidopsis, maize and rice [33,34,35,36]. Besides the type of SNP variation, it is also important to know the location of the variation in the genome, for example whether the SNP is in the coding region, regulatory region, or other part of the genome, and how the variation affects functional genes and phenotypic expression [37]. The SNPs could affect the process of gene expression [38] and thereby be responsible for phenotypic variation (e.g., stress tolerance/resistance, yield performance, etc.) among the genotypes. The generated SNP markers could also be useful in future genomic studies to study trait association, determine the genome location of the SNPs, identify diagnostic markers for selection breeding and to understand their effect on the molecular processes (e.g., gene expression) responsible for phenotypic traits of interests (e.g., drought and salt tolerance) in Rhodes grass and related grass species. 

### 4.2. Genome Wide Mapping of Markers

In genomic studies, a reference genome is considered very important to select genome wide markers for genetic diversity and population structure analyses as well as to study markers linked with genes regulating traits of agronomic importance. Following recent advances in the genomic toolbox, reference genomes have been developed for many crops [39,40,41]. However, a lack of reference genomes is one of the main challenges in the genomic studies of many orphan crops which include most tropical forages. Due to this limitation, we assessed the opportunity to use reference genomes of closely related species to advance our genomic studies. Taxonomic classification was used as the criteria to select between the candidate reference genomes for Rhodes grass. Similar approaches have been used to select reference genomes for Buffelgrass and Napier grass [42,43], which enabled the selection of representative sets of genome wide markers for diversity and population structure analyses. Accordingly, we initially selected the reference genomes of tef [28], finger millet [30] and Manila grass [29] to map the generated markers with the aim of selecting genome wide representative markers for in-depth diversity and population structure analyses. However, only a small proportion of the generated DArTSeq markers were able to be mapped onto the selected reference genomes. In an effort to identify the genomic position of more markers, we also attempted to map the generated markers onto the reference genome of foxtail millet [31,32], but the number of mapped markers was low, even compared to the above reference genomes. The low number of mapped markers could be attributed to the limited genomic information available from the reference genomes, resulting in the mapping of only a small proportion of the generated markers. The other reason could be the existence of a weak level of synteny between the genome of Rhodes grass and the selected reference genomes. Similar results were reported in Buffelgrass and Napier grass using reference genomes of closely related species. In buffelgrass, only 12% of the SNP markers were mapped using the *Setaria italica* reference genome [42]. In Napier grass, initially only 17% and 33% of the SilicoDArT and SNP markers, respectively, were mapped using the pearl millet genome [43]. However, more than 80% of the markers were subsequently mapped onto the Napier grass genome when it became available [44]. In addition, except for *Setaria italica*, the reference genomes were not assembled at the chromosome level. This has hindered the effort of mapping the markers onto known genomic positions in the reference genomes. As a result, we were not able to select genome wide mapped representative markers for further genetic analyses such as LD and LD decay. Due to this challenge, we chose markers for diversity analysis and population structure based on the loading score obtained by DAPC analysis and the PIC values. In the future, it will be useful to generate a reference genome for Rhodes grass which will enhance the opportunity to use genomic technologies to support the further development of this forage species, including resolving the challenge of selecting sufficient genome wide markers for in-depth diversity, LD, LD decay and QTL identification studies. 

### 4.3. Genetic Diversity and Population Structure 

The generated markers were used to study the genetic diversity and population structure of the Rhodes grass collection. The results revealed the presence of a broad range of diversity in the collection, as demonstrated by the hierarchical clustering as well as Bayesian population structure analysis. In the hierarchical analyses, two and three clusters were observed using SNP and SilicoDArT markers, respectively, with an 82% cophenetic correlation coefficient between the two marker types. In the case of the Bayesian population structure analysis, the delta K suggested the presence of two subpopulations in the collection using both marker types. The AMOVA result supported the clustering and population structure by partitioning the total variation into among and within clusters and there was a significant difference in terms of the accessions in the clusters and subpopulations identified using cluster and population structure analyses, respectively. There is no evaluation data, across locations and years, for the collection to relate with the result of the current study. We have considered the available characterization data and the passport data of the accessions (whenever possible), which is provided as Appendix A to provide readers with information about the geographic origin of the accessions. However, there is no clear signature for origin-based grouping in the collection as accessions from the same country were distributed in different clusters. This is supported by the low Mantel correlation coefficient between the genetic distance and geographical distance. In terms of genetic diversity, the results of the current study are broadly in line with a previous report, based on agronomic and morphological traits, which indicated the rich diversity in the collection as revealed by grouping of 62 accessions into six clusters [15]. The agronomic traits that classified the accessions included plant height, yield and the percentage of leaves to total biomass, while the morphological traits included plant height, growth habit, stolons, internode length, leaf length and width, hairiness of leaf, ligule and awn, flowering characters and culm thickness [15]. However, we did not identify a specific complementary between our genotypic data and passport data, as there was no clear correlation between the agro-morphological and molecular analyses, with accessions from the different groups identified using agro-morphological traits distributed among the clusters identified using the molecular data.

There are only a few reports on the application of conventional molecular markers to Rhodes grass genetic studies. ISSR, SRAP and AFLP markers have been used to study diploid and tetraploid cultivars [9,13]. A genetic linkage map of Rhodes grass was developed using AFLP and RFLP markers [12]. However, in most cases, low density markers were used to differentiate a small number of known cultivars [8,9,10,11,12,13]. In addition, as only a few cultivars were included, the scope of these studies was limited to inferring how the results would apply to the wider genetic diversity and population structure in the Rhodes grass germplasm from around the world. In-depth genetic diversity, population structure and genomic studies are limited in the species. Thus, we leveraged the recent development in the genotyping technologies to generate high density marker sets to study a relatively large number of accessions in our collection. In the current study, 104 accessions, mostly from different African countries, were included, and the observed clusters demonstrated the genetic diversity held in the collection. However, although one of the largest collections, there is a substantial gap in the ILRI collection, as it lacks germplasm from other countries where Rhodes grass is native. Some of the African countries where Rhodes grass is native but not represented in the in trust collection include Angola, Benin, Botswana, Cameroon, Chad, Djibouti, Egypt, Eritrea, Gambia, Ghana, Lesotho, Mali, Morocco, Mozambique, Namibia, Niger, Nigeria, Somalia and Tunisia [2,3]. The crop is also native to Bahrain, Oman, and Saudi Arabia in Asia, and to Portugal and Spain in Europe [2,3]. Germplasm from these countries could provide a distinct addition to broaden the diversity of this collection. A gap analysis is yet to be done to quantify the level of the species’ diversity maintained in the collection. Hence, the generated markers could be used to identify any gap and enable the introduction of potentially distinct materials into the collection. Besides providing insights into the genetic variation contained within our collection, the generated marker data could be used to study the genetic diversity of Rhodes grass collections maintained in other centers (such as GeRRI (Kenya), USDA and ICBA (UAE)) to identify unique material or duplicates among the collections, to improve our understanding and ability to exploit the genetic diversity that is contained within these collections and to develop a representative core collection for the global germplasm held in the different centers in the future. In general, this study has generated large sets of markers as well as useful genetic information on a relatively large number of Rhodes grass accessions. 

### 4.4. Representative Subset Development 

Besides the lack of information on diversity, evaluation of forage crops, over harvests and years, is an expensive and labor-intensive initiative to use the whole ILRI collection for research and development to, for example, select promising lines for different agro-ecologies. Moreover, resources (space, funding, etc.) for tropical forages are limited. This challenge can be managed by developing a subset that is representative of the full genetic diversity held in the collection. Taking this challenge into account, we used the generated SNP markers to develop a representative subset, containing 20% of the total number of accessions in the collection. The subset contained 21 accessions, which originated from various countries. The result of the AMOVA showed that there was no significant genetic variation between the developed subset and the whole of the collection. Thus, the developed subset will provide an opportunity to use a representative set of the collection for future improvement of Rhodes grass. For the example, the subset can be used as an entry to the whole collection to evaluate a broad range of the diversity of the collection under different agro-ecologies, which could enhance the use of the rich diversity held in the ILRI Genebank and thereby contribute to the improvement of smallholder farmers’ livelihoods, which are dependent on livestock production in the tropical and subtropical environments. 

## 5. Conclusions

High throughput DArTSeq markers were generated and used for genetic diversity and population structure analyses of a Rhodes grass collection held in the ILRI genebank. The results revealed the rich genetic diversity held in the collection as demonstrated by cluster and structure analyses. A subset, representative of the collection’s diversity, was developed which could enhance the use of the collection for research and development and support the selection of improved climate resilient Rhodes grass varieties for different agro-ecologies. In general, we strongly believe the observed diversity and the developed subset will be a catalyst for further research to evaluate and identify genotypes for climate resilient farming, particularly in drought and saline affected environment, in this crop. 

## Figures and Tables

**Figure 1 genes-12-01233-f001:**
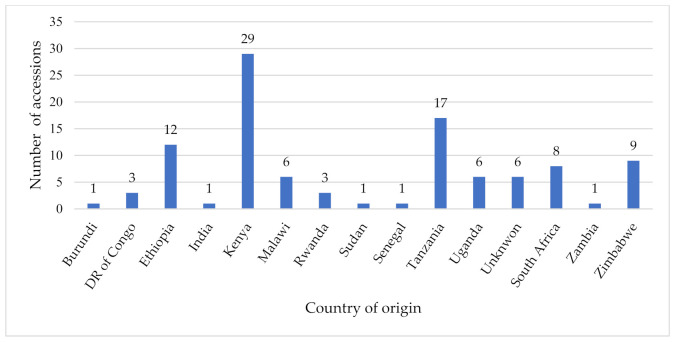
Rhodes grass accessions assessed by country of origin.

**Figure 2 genes-12-01233-f002:**
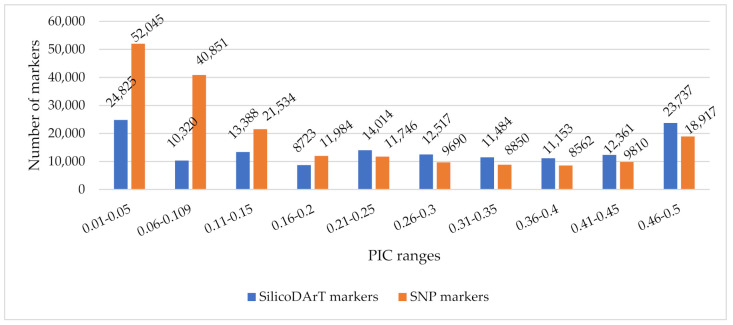
Number of markers by PIC values.

**Figure 3 genes-12-01233-f003:**
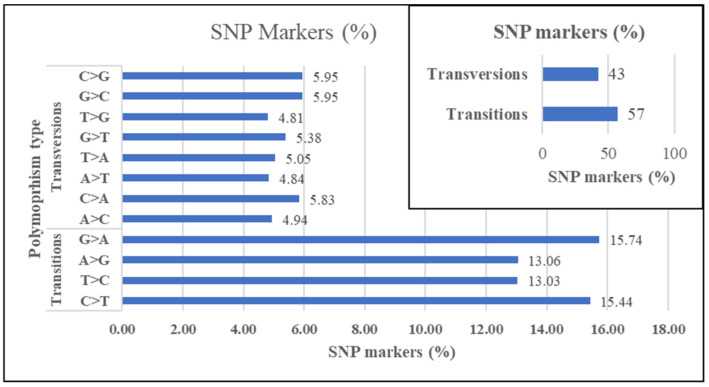
Proportion of SNP markers by transition and transversion polymorphisms.

**Figure 4 genes-12-01233-f004:**
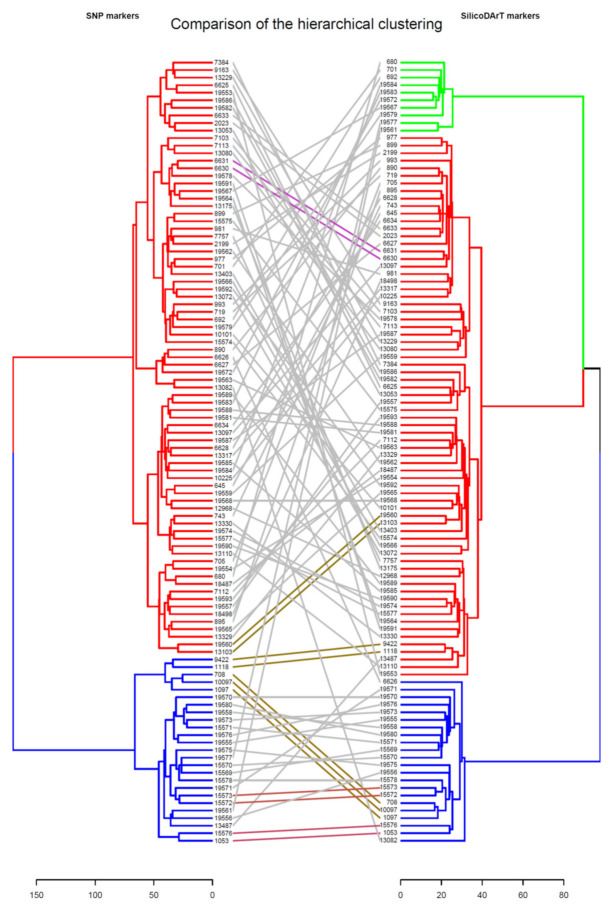
Hierarchical clustering of Rhodes grass accessions using SNP and SilicoDArT markers.

**Figure 5 genes-12-01233-f005:**
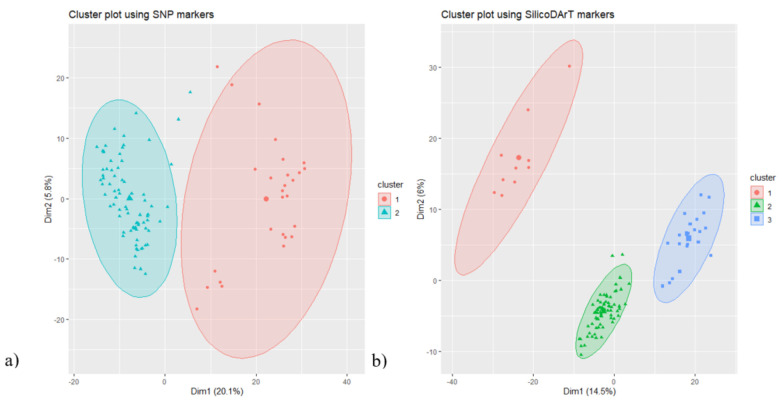
Cluster plots showing population clusters using (**a**) SNP and (**b**) SilicoDArT markers.

**Figure 6 genes-12-01233-f006:**
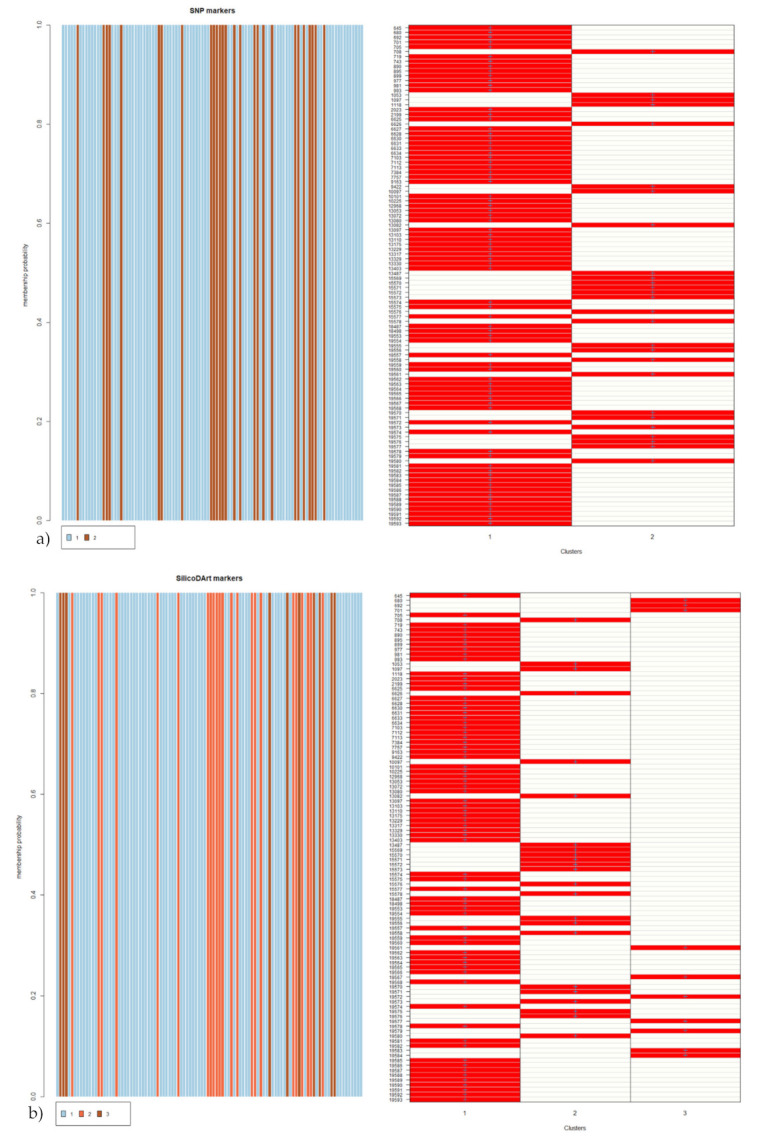
Accessions cluster membership probability and assignment of accessions to each cluster based on (**a**) SNP and (**b**) SilicoDArT markers.

**Figure 7 genes-12-01233-f007:**
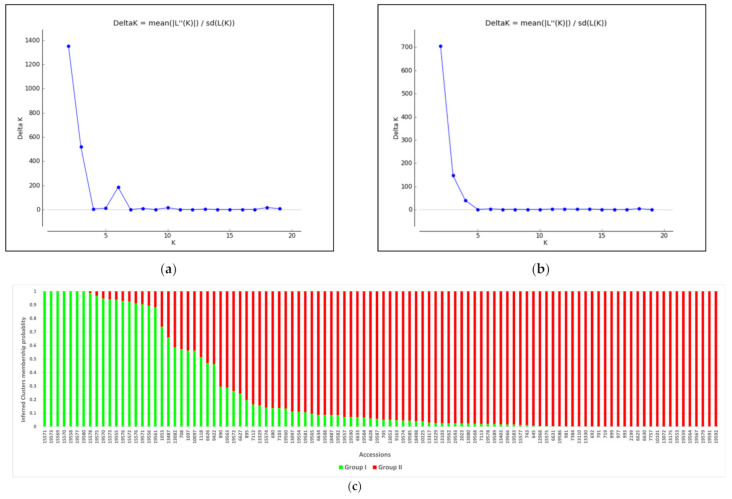
Bayesian assignment of population structure of 104 Rhodes grass accessions: Delta K (ΔK) for K = 2 to 20 subpopulations using SNP (**a**) and SilicoDArT (**b**) markers. The STRUCTURE barplot of the estimated membership coefficient (Q) of each accession for K = 2 for SNP (**c**) and SilicoDArT (**d**) markers. Each bar represents the Q of an individual accession.

**Table 1 genes-12-01233-t001:** Number of markers mapped onto the different reference genomes.

Reference Genomes	Number of Markers Mapped		Remark
	SilicoDArT *	SNP *	
Finger millet (*Eleusine coracana*)	1518 (0.8%)	7846 (5.5%)	Scaffolds
Tef (*Eragrostis tef*)	986 (0.5%)	4008 (2.8%)	Scaffolds
Manila grass (*Zoysia matrella*)	817 (0.4%)	3554 (2.5%)	Scaffolds
Foxtail millet *(Setaria italica)*	212 (0.1%)	1354 (1%)	Chromosomes, scaffolds, and plastid

* Number in parentheses is the percentage of mapped markers.

**Table 2 genes-12-01233-t002:** Analysis of molecular variance (AMOVA) of clusters identified using high throughput DArTSeq markers.

Marker Type	Source of Variation	Degree of Freedom	Sum of Square	Mean Sum of Square	Estimation Variation	Percentage of Variation	PhiPT	*p* Values
SNP	Among clusters	1	4094.732	4094.732	97.418	33%	0.328	0.000
	Within clusters	102	20,389.393	199.896	199.896	67%		
	Total	103	24,484.125		297.314	100%		
SilicoDArT	Among clusters	2	4595.581	2297.791	84.988	30%	0.303	0.000
	Within clusters	101	19,711.217	195.161	195.161	70%		
	Total	103	24,306.798		280.148	100%		

**Table 3 genes-12-01233-t003:** Analysis of molecular variance (AMOVA) of the genetic variation among and within the of Rhodes grass subpopulations collection using high throughput DArTSeq markers.

Marker Type	Source of Variation	Degree of Freedom	Sum of Square	Mean Sum of Square	Estimation Variation	Percentage of Variation	PhiPT	*p* Values
SNP	Among subpopulation	1	3888.551	3888.551	86.352	30%	0.299	0.000
	Within subpopulation	102	20,601.238	201.973	201.973	70%		
	Total	103	24,489.788		288.325	100%		
SilicoDArT	Among subpopulation	1	2309.165	2309.165	51.157	19%	0.192	0.000
	Within subpopulation	102	21,997.633	215.663	215.663	81%		
	Total	103	24,306.798		266.820	100%		

**Table 4 genes-12-01233-t004:** List of accessions, origin, and cluster groups contained in subset developed using SNP markers.

Accession #	Doi	Cluster	Origin
680	10.18730/G6099	I	Tanzania
890	10.18730/G7KAE	I	Tanzania
895	10.18730/G7KFK	I	Tanzania
1118	10.18730/FQ2D=	II	Congo
6627	10.18730/G5WP5	I	South Africa
6628	10.18730/G5WQ6	I	Unknown
6633	10.18730/G5WVA	I	Tanzania
6634	10.18730/G5WWB	I	Unknown
10097	10.18730/FP5R6	II	Ethiopia
13487	10.18730/FS6BT	II	Ethiopia
15576	10.18730/FTTPU	II	Ethiopia
19554	10.18730/FYAEQ	I	Tanzania
19557	10.18730/FYAHT	I	Kenya
19558	10.18730/FYAJV	II	Congo
19563	10.18730/FYAQ *	I	Zimbabwe
19568	10.18730/FYAW0	II	Kenya
19572	10.18730/FYB04	I	India
19581	10.18730/FYB9D	I	Tanzania
19583	10.18730/FYBBF	I	Malawi
19584	10.18730/FYBCG	I	Kenya
19590	10.18730/FYBJP	I	Kenya

**Table 5 genes-12-01233-t005:** AMOVA result between the subsets and the rest of the population.

Marker Type	Source of Variation	Degree of Freedom	Sum of Square	Mean Sum of Square	Estimation Variation	Percentage of Variation	PhiPT	*p* Values
SNP	Between groups	1	245.363	245.363	0.229	0.10%	0.001	0.311
	Within groups	102	24,244.425	237.690	237.690	99.90%		
	Total	103	24,489.788		237.919	100.00%		

## Data Availability

All data generated in this study are freely available as international public goods.

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
