# Peer review of "Genetic Diversity and Population Structure of a Rhodes Grass (Chloris gayana) Collection"

_genes, 2021, doi:10.3390/genes12081233_

Round 1

Reviewer 1 Report

Comments:

Introduction - short description of Rhodes grass, its agronomic importance, second paragraph just describing the markers used in previous studies, lack a better and detailed overview of this issue (which is the topic and aim of the study), then just follows the description of collections (but these are not used in any way in the study)

Materials and methods - it is described that 104 accessions from genebank have been used, why is it mentioned that leaf samples for two accessions have not been obtained? (which), 104 samples are listed in the table; it is mentioned in the introduction (line 66) that the collection contains wild accessions (or landraces?) and cultivars - there is no indication in the sample list (what samples are cultivars); the sample set is narrow enough to affect variability of the species within the area of occurrence; why samples from other collections have not been included?; poor  no description of the selection of reference genomes and mapping markers to reference genomes; missing description of R-packages used for calculations, missing functions in line 106

Results - Figure 2 - redundant, a number of markers have scant telling value, is not commented; the selection of reference genomes for marker mapping is strange - figure 4 does not represent a phylogenetic relationship, it is just a simple taxonomic classification, the erroneous selection of reference genomes is also underlined by the results when the most markers have been mapped into the genome of Eleusine corocana (according to the table and commentary info a species that has a different number of chromosomes, is more distant and was not the favourite for selection); it is not clear that the detailed analyses relate to a selected 1000 markers, figure 5 is also redundant; the results of hierarchical clustering are interesting but virtually uncommented and unexplained, the description of the number of samples from each country is meaningless, no comment is made on the difference between cluster (position of samples in clusters) for SNP and DArT, no good explanation is given for figure 6 - links between markers/clusters, no comment on whether or not there is similarity between samples from one country/geographical region; no comments on  8, figure 7 is redundant; no comment on figure 9; population structure - delta K images are redundant

Discussion - the link between the SNP and the phenotype is discussed but not supported by data, just as the location of the SNP on chromosomes is a hypothetical matter in this case; the link to agronomic features is discussed - but then it is stated that there was no correlation, it is not explained, it is not explained whether there was difference between cultivars and wild genotypes...

What's beneficial is the definition and selection  of a core collection of genotypes.

Author Response

Dear Editors and Reviewer, 

With regards,

Alemayehu Teressa Negawo

Reviewer 2 Report

Genotyping tropical forages such as the Rhodes grass is difficult but useful and necessary work. The authors identified extensive genetic diversity and generated valuable data. But I was surprised to see the analysis did not really find any explanations for the genetic diversity: the genetic clusters did not correlate with each other, or with morphology, and apparently did not show any geographical signal either. Most grasses like these show a strong geographical signal in their genetic diversity, so I am surprised this was not the case here – was geographical mapping attempted? My biggest recommendation is that the authors review their data again to try to bring it all together into a logical overarching structure.

The writing begins fluently and confidently in the introduction, but the narrative in the discussion has lower quality writing and does not make logical sense all the way through; I recommend some text work to streamline the narrative.

Introduction and Table s1:

I was surprised by the lack of collection history presented. How dos ILRI come to have these collections? Those data should be included in table S1.

Have herbarium vouchers been made of any of these accessions, and deposited to herbaria? That would be very to know also if digitised online records and/or images were available.

Results:

Please refer to the NCBI genomes by their reference numbers, not URLs

If the Oropetium genome was not used, do not mention it – or explain why it was not used

Figure 4 is not necessary, please delete. It does not show any phylogenetic relationships

Figure 9: the writing is too small to be visible

Discussion:

It is not necessary to explain transition/transversion in a journal paper, please remove

I very much hope the data generated will be uploaded to an online repository, but I did not see this clearly stated.

Author Response

(The authors gave the same response as above.)

Round 2

Reviewer 2 Report

Thank you